# Cathepsin B and Plasma Kallikrein Are Reliable Biomarkers to Discriminate Clinically Significant Hepatic Fibrosis in Patients with Chronic Hepatitis-C Infection

**DOI:** 10.3390/microorganisms10091769

**Published:** 2022-09-01

**Authors:** Alexia de Cassia Oliveira Zanelatto, Gilmar de Souza Lacerda, Camila de Melo Accardo, Natalia Fonseca do Rosário, Andréa Alice da Silva, Guacyara Motta, Ivarne Luis dos Santos Tersariol, Analucia Rampazzo Xavier

**Affiliations:** 1Departamento de Bioquímica, Escola Paulista de Medicina, Universidade Federal de São Paulo, São Paulo 04044-020, SP, Brazil; 2Laboratório Multiusuário de Apoio à Pesquisa em Nefrologia e Ciências Médicas, Departamento de Medicina Clínica—LAMAP, Faculdade de Medicina, Universidade Federal Fluminense, Niterói 24033-900, RJ, Brazil; 3Departamento de Patologia, Faculdade de Medicina, Universidade Federal Fluminense, Niterói 24033-900, RJ, Brazil

**Keywords:** HCV, hepatic fibrosis, chronic hepatitis C, plasma kallikrein, cathepsin B, TGF-β

## Abstract

We aimed to determine the biomarker performance of the proteolytic enzymes cathepsin B (Cat B) and plasma kallikrein (PKa) and transforming growth factor (TGF)-β to detect hepatic fibrosis (HF) in chronic hepatitis C (CHC) patients. We studied 53 CHC patients and 71 healthy controls (HCs). Hepatic-disease stage was determined by liver biopsies, aminotransferase:platelet ratio index (APRI) and Fibrosis (FIB)4. Hepatic inflammation and HF in CHC patients were stratified using the METAVIR scoring system. Cat-B and PKa activities were monitored fluorometrically. Serum levels of TGF-β (total and its active form) were determined using ELISA-like fluorometric methods. Increased serum levels of Cat B and PKa were found (*p* < 0.0001) in CHC patients with clinically significant HF and hepatic inflammation compared with HCs. Levels of total TGF-β (*p* < 0.0001) and active TGF-β (*p* < 0.001) were increased in CHC patients compared with HCs. Cat-B levels correlated strongly with PKa levels (r = 0.903, *p* < 0.0001) in CHC patients but did not correlate in HCs. Levels of Cat B, PKa and active TGF-β increased with the METAVIR stage of HF. Based on analyses of receiver operating characteristic (ROC) curves, Cat B and PKa showed high diagnostic accuracy (area under ROC = 0.99 ± 0.02 and 0.991 ± 0.007, respectively) for distinguishing HF in CHC patients from HCs. Taken together, Cat B and PKa could be used as circulating biomarkers to detect HF in HCV-infected patients.

## 1. Introduction

Hepatitis-C virus (HCV) infection is the most common blood-borne infection worldwide. HCV infection is associated with an increased risk of hepatic fibrosis/cirrhosis and hepatocellular carcinoma and is the major indication for liver transplantation [1,2]. An estimated 71.1 million people are infected with HCV worldwide and ~700,000 in Brazil [1,3]: HCV infection is a serious public-health issue.

HCV is highly dependent on the host’s lipid metabolism to create a favorable environment for its replication in the liver [4,5,6,7]. Internalization of HCV into hepatocytes occurs via clathrin-mediated endocytosis, and the low pH of the endosomal compartment induces the fusion and replication of HCV [8]. HCV induces cytokine production and antibodies through the activation of the immune system. The latter eliminates HCV-infected cells through direct cytotoxic or non-cytolytic mechanisms [9]. HCV infection can promote hepatic stellate cell (HSC) activation by external stimuli. Consequently, several specific phenotypic changes occur that distort the hepatic architecture: proliferation, contractility, fibrogenesis, altered matrix degradation, chemotaxis and inflammatory signaling [10].

Hepatic fibrogenesis is mediated by proteases secreted by hepatocytes, HSCs and endothelial cells [11]. Matrix metalloproteinases, kallikreins and cysteine cathepsins, such as cathepsin B (Cat B) and Cat L, can degrade components of the extracellular matrix by proteolytic means. Therefore, proteolytic enzymes play a critical role in the pathogenesis of hepatic fibrosis. Once the molecules involved directly in the formation of fibrotic scars and regenerative nodules are activated, progression to cirrhosis can occur [11,12]. Activation of protease-activated receptor-2 by proteolytic enzymes increases production of transforming growth factor (TGF)-β, and induces a profibrogenic phenotype in human HSCs and promotes hepatic fibrosis in mice [13].

The treatment of HCV infection is dependent upon clinical and laboratory information: the staging of liver damage, the viral genotype, and the presence of coinfections [3]. The Hepatic Fibrosis Score is determined by liver biopsies (“gold standard”), hepatic elastography, or can be estimated by the mathematical calculations aminotransferase:platelet ratio index (APRI) and Fibrosis (FIB)4 (which are more efficient only in advanced stages of hepatic diseases) [14,15]. Liver biopsy is an invasive procedure with a bleeding risk for patients with hepatic fibrosis. Elastography has been increasingly used by clinicians for the exclusion of liver damage in normal elastography results in patients without any defined risk factors rule out or diagnoses of clinically significant fibrosis, while abnormal results in patients at risk confirm diagnoses of advanced fibrosis or cirrhosis. However, in patients with an intermediate risk of hepatic fibrosis, additional tests are needed to establish an accurate diagnosis [15,16].

Scholars have searched for serologic markers that may show altered hepatic function or be predictive of the severity of hepatic fibrosis [17]. In this scenario, cathepsins and other proteases emerge as possible biomarkers for the estimation of hepatic lesions [12].

We conducted a study in patients with chronic hepatitis-C (CHC) infection to investigate the diagnostic performance of the serum levels of Cat B and plasma kallikrein (PKa) for predicting hepatic fibrosis.

## 2. Materials and Methods

### 2.1. Ethical Approval of the Study Protocol

The study protocol was in accordance with the ethical guidelines of the 1975 Declaration of Helsinki. It was approved by the Research Ethics Committee of Universidade Federal Fluminense (Niterói, RJ, Brazil) (35033514.5.0000.5243) and Universidade Federal de São Paulo (City of São Paulo, SP, Brasil) (72891317.9.0000.5505).

### 2.2. Exclusion Criteria

The exclusion criteria were patients: with co-infections; with hepatocellular carcinoma (or a history of it); with diabetes mellitus; using drugs that could modify lipid/glucose metabolism; with another chronic liver disease, with non-alcoholic fatty liver disease (NAFLD), and with non-alcoholic steatohepatitis (NASH); with altered blood parameters; addicted to alcohol; and who were pregnant or lactating.

### 2.3. Recruitment

CHC patients were recruited from the Hepatitis Treatment Center at Hospital Universitário Antônio Pedro (Niterói, RJ, Brazil). Healthy controls (HCs) and healthy blood donors were recruited from the Colsan Foundation (São Paulo, SP, Brazil).

### 2.4. Cohorts

Seventy-one HCs were recruited from the Colsan Foundation, and 53 CHC patients who underwent liver biopsy and laboratory parameter testing from March 2015 to January 2019 were considered for inclusion.

The liver biopsy results of the CHC patients were based on the METAVIR scoring system [3]. The grade indicates inflammatory activity (A0: none; A1: mild; A2: moderate; A3: severe), whereas the stage represents the extent of fibrosis/scarring (F0: no fibrosis; F1: portal fibrosis without septa; F2: portal fibrosis with few septa; F3: numerous septa without cirrhosis; F4: cirrhosis).

The viral genotype was obtained from the medical records of the patients.

We evaluated the laboratory parameters of liver function (aspartate aminotransferase (AST), alanine aminotransferase (ALT), total bilirubin (TB) and fractions (direct (DB) and indirect (IB), alkaline phosphatase (ALP), gamma glutamyl transferase (GGT), coagulation tests) and hematologic parameters of CHC patients. TGF-β (total and active), Cat B and PKa were measured in CHC patients and HCs. APRI and FIB4 scores were calculated using mathematic formulae [18,19]: APRI = (AST [level]/AST [upper limit of normal]/platelet count [10^9^/L]) × 100. FIB4 = age [years] × (AST [IU/L]/platelet count [expressed as platelets × 109/L] × ALT1/2 [IU/L]). The cutoff for APRI (≥1.5) and FIB4 (3.25) was used in analyses of receiver operating characteristic (ROC) curves to ascertain the presence of HF in HCV infection [18,19,20].

Subjects from the control group were blood donors, free of viral infections and hepatic commitment, or another disease clinically detected. All CHC patients, including the cirrhotic group, were clinically compensated at the time of this study under outpatient medical follow-up. None of the CHC patients had previously experimented with first- and second-generation antiviral drugs.

### 2.5. Blood Sampling

Samples of venous blood were collected in the morning after a 12-h overnight fast in Vacutainer^®^ blood-collection tubes (Becton Dickson, Franklin Lakes, NJ, USA). These tubes contained a serum-clotting activator (to obtain serum) and 3.2% sodium citrate or ETDA-K_2_ (to obtain plasma and total blood). Samples were processed immediately and, if necessary, were stored at −80 °C for analyses.

### 2.6. Laboratory Assays

Serum levels of AST, ALT, ALP, GGT, TB, DB and IB were measured in a RxL Max™ Clinical Chemistry System (Siemens Healthcare Diagnostics, Newark, DE, USA). Hematologic parameters were assessed using a LH750^®^ system (Beckman Coulter, Fullerton, CA, USA). Plasma samples were used to measure prothrombin time and partial thromboplastin time using the CA-1500 system (Sysmex, Lincolnshire, IL, USA). The International Normalized Ratio was also calculated.

Serum levels of total TGF-β and its active form (active TGF-β) were measured using enzyme-linked immunosorbent assay (ELISA) kits in 96-well clear plates according to manufacturer (Invitrogen, Carlsbad, CA, USA) instructions. Absorbance at 450 nm and 570 nm in plates was measured using a spectrophotometer (Spectramax™ M3; Molecular Devices, Silicon Valley, CA, USA) and analyzed using SoftMax™ Pro 6.0 (Molecular Devices). The samples were tested in triplicate.

Cat B and PKa endopeptidase activities in the serum of patients and HCs were measured by hydrolysis of the fluorogenic substrate benzyloxycarbonyl-L-phenylalanyl-L-Arginine 4-Methyl-Coumaryl-7-Amide (50 µM; Bachem Americas, Torrance, CA, USA). The mean fluorescence intensity was quantified using a FlexStation^®^3 microplate reader (Molecular Devices, San Jose, CA, USA). The excitation wavelength and emission wavelength were set at 380 nm and 460 nm, respectively.

To verify the specificity of proteolytic enzymes, we carried out assays in the presence or absence of proteinase inhibitors. We used 10 µM of the Cat B-specific inhibitor L-3-trans-(Propylcarbamoyl)oxirane-2-carbonyl]-L-isoleucyl-L-proline (catalog number CA-074; Sigma-Aldrich, St. Louis, MO, USA) and 20 µM of the PKa-selective inhibitor *trans*-4-Aminomethyl cyclohexane carbonyl-phenylalanyl-4 carboxymethyl anilide (catalog number PKSI-527; Santa Cruz Biotechnology, Santa Cruz, CA, USA) [21,22]. The samples were tested in triplicate.

### 2.7. Statistical Analyses

Groups were compared using the Mann–Whitney *U*-test or Student’s *t*-test, as appropriate. Comparisons between the score of indices (APRI, FIB4) and the mean level of biomarkers (Cat B and PKa) were undertaken using a one-way analysis of variance with Tukey’s multiple comparisons test using each group as a fixed factor. The correlation between serum levels of Cat B and PKa was analyzed using Pearson’s correlation coefficient. The diagnostic performance of individual biomarkers was investigated using the area under the receiver-operating characteristics curve (AUROC) with a 95% confidence interval (CI). Sensitivity and specificity were determined for appropriate cutoff values based on the ROC curves. Results are the mean ± SEM with 95%CI unless stated otherwise. *p* < 0.05 was considered significant. Statistical analyses were undertaken using Prism 6.0 (GraphPad, San Diego, CA, USA).

## 3. Results

### 3.1. Baseline Characteristics of CHC Patients

The baseline characteristics of the CHC patients are summarized in Table 1. Viral genotype 1 was the most prevalent, followed by viral genotype 3. The clinical laboratory parameters of all CHC patients revealed no jaundice (with bilirubin values within reference ranges) and showed alterations in laboratory values compatible with CHC infection. Distributions for fibrosis stage and inflammatory activity grade in the study population are also presented in Table 1. If the mean APRI score was >1.5, CHC patients had hepatic cirrhosis. As expected, the HCs had no hepatic fibrosis. FIB4 scores in the CHC group suggested advanced HF (>3.25). Normal APRI and FIB4 scores were found in the HCs.

CHC patients were clinically stable, as reinforced by the mean of the laboratorial parameters. No encephalopathy is observed, severe coagulation disorders, or the presence of evident ascites. Patients with cirrhosis detected on liver biopsy (F4, N = 13) had a few points in the Child–Pugh criterion, which can be classified as class A (6/13).

### 3.2. Serum Levels of Cat B and PKa Increased with the METAVIR Stages in HCV Patients

Patients were stratified into groups according to the degree of hepatic fibrosis (F2, F3 or F4) as presented by liver biopsy (Figure 1). APRI and FIB4 scores (Figure 1A,B) could be used to stratify the advanced stages of hepatic fibrosis/cirrhosis (F4 versus F3, F2 and HCs, *p* < 0.05), but, as expected, significant results were not found when comparing F3 versus F2 or F2 versus HCs. However, serum levels of Cat B and PKa (Figure 1C,D) could be used to separate HCs from F2, F2 from F3, and F3 from F4. These data strongly suggested that serum levels of Cat B and PKa could provide clinically relevant diagnostic accuracy as a single marker of significant (≥F2), advanced (≥F3) hepatic fibrosis and hepatic cirrhosis (≥F4). When CHC patients were stratified according to hepatic inflammation (Figure 1E,F), serum levels of Cat B and PKa could be used to separate HCs from A1, A1 from A2, but could not be used to separate A2 from A3.

### 3.3. Evaluation of the Diagnostic Performance of Cat B and PKa to Detect Liver Fibrosis

The diagnostic performance of APRI, FIB4 and levels of Cat B and PKa to detect hepatic fibrosis are depicted in Figure 2. Analyses of the ROC curves demonstrated the diagnostic power of APRI (Figure 2A). The APRI AUROC was 0.71 ± 0.05 (95%CI = 0.621–0.807; *p* < 0.0001). This result showed that APRI had only a moderate diagnostic capacity to detect hepatic fibrosis ≥ A1F2 in CHC patients. APRI ≥ 1.5 showed that 14 of 53 CHC patients were correctly identified as having hepatic fibrosis (sensitivity = 26%); three HCs were identified as false-positive (4.2%). APRI < 1.5 correctly identified 68 HCs who were negative for hepatic fibrosis (specificity = 96%); 39 CHC patients with hepatic fibrosis ≥ A1F2 were detected as false-negative (73%). The prevalence of 2.7% liver fibrosis [23] was used to estimate the negative value (NPV) = 97.9%. This result showed that APRI had a moderate capacity to include CHC patients with hepatic fibrosis (accuracy = 66.1%). APRI could not be used to discriminate clinically significant hepatic fibrosis (≥F2) in CHC patients from HCs, which confirmed our previous results (Figure 1A).

The diagnostic power of FIB4 (Figure 2B) was also low (AUROC = 0.67 ± 0.06; 95%CI = 0.525–0.817; *p* < 0.0001; N = 124). A FIB4 score ≥ 3.25 showed that 22 CHC patients were correctly identified as having hepatic fibrosis (sensitivity = 41%); seven HCs were identified as false-positive for hepatic fibrosis (9.9%). A FIB4 score < 3.25 could be used to correctly identify 64 HCs (specificity = 90%) who were negative for hepatic fibrosis (NPV = 98.2%); 31 CHC patients with hepatic fibrosis ≥A1F2 were detected as negative (58.5%).

The diagnostic performance of the Cat B level (Figure 2C) to detect clinically significant hepatic fibrosis (≥F2) in CHC patients was very high (AUROC = 0.992 ± 0.006; 95%CI = 0.9803–1.003; *p* < 0.0001). The cutoff values of Cat B ≥2.61 U/mL showed that 51 of 53 CHC patients were identified correctly (sensitivity = 96%) with clinically significant hepatic fibrosis (≥F2); three HCs were identified as false-positive (4.2%). A serum level of Cat B < 2.61 U/mL correctly identified 68 HCs as negative for hepatic fibrosis (specificity = 96%); two CHC patients were detected as false-negative (3.8%), showing NPV = 99.9%. This test was reliable for the identification of hepatic fibrosis (≥A1F2) in previously screened CHC patients; the high prevalence = 85% of liver fibrosis [1] in patients infected with HCV was used to estimate the positive predictive value (PPV) = 99%.

The serum level of PKa (Figure 2D) displayed high diagnostic power to detect clinically significant hepatic fibrosis in CHC patients from HCs (AUROC = 0.991 ± 0.007; 95%CI = 0.9770–1.005; *p* < 0.0001; N = 124). Using a cutoff value of PKa ≥ 2.9 U/mL, this test could be used to correctly identify 51 of 53 CHC patients (sensitivity = 96%) with clinically significant hepatic fibrosis (≥F2); one HC was identified as false-positive (1.4%). PKa < 2.9 U/mL correctly identified 70 HCs as negative for hepatic fibrosis (specificity = 98%); two CHC patients were false-negative (3.8%), showing NPV = 99.9%. This test was reliable for identifying hepatic fibrosis (≥A1F2) in previously screened CHC patients; the high prevalence = 85% of liver fibrosis [1] in patients infected with HCV was used to estimate the positive predictive value (PPV) = 98%.

The AUROC of cathepsin B was also calculated for each grade of the liver fibrosis. We observed the diagnostic power of cathepsin B to predict liver fibrosis as a function of the METAVIR score: Control vs. F2 + F3 + F4, AUROC = 0.992 ± 0.006 (95%CI = 0.98–1.00; *p* < 0.0001), Control + F2 vs. F3 + F4 AUROC = 0.98 ± 0.01 (95%CI = 0.96–1.01; *p* < 0.0001) and Control + F2 + F3 vs. cirrothic F4 AUROC = 0.95 ± 0.02 (95%CI = 0.91–0.99; *p* < 0.0001). We also determined the diagnostic power of cathepsin B in F1 patients, who were not previously classified as CHC patients. Cathepsin B only displayed a moderate diagnostic capacity to detect fibrosis in F1 patients, AUROC = 0.68 ± 0.09 (95%CI = 0.51–0.86; *p* = 0.035), but cathepsin B also showed excellent diagnostic power to detect intermediate levels of fibrosis in F2 patients (AUROC = 0.992 ± 0.006) and advanced liver fibrosis in F3 (AUROC = 0.98 ± 0.01) and F4 patients (AUROC = 0.95 ± 0.02).

### 3.4. Serum Levels of Cat B and PKa Are Correlated in CHC Patients

Figure 3A shows the correlation between the levels of Cat B and PKa in the serum of CHC patients with fibrosis ≥A1F2. A strong positive correlation was observed between these variables (r = 0.903, 95%CI = 0.837–0.943, *p* < 0.0001). However, this same correlation was not observed in the HCs (Figure 3B). These results strongly suggest that the serum levels of Cat B and PKa were associated with hepatic fibrosis in CHC patients.

### 3.5. Association between Cat B and the Active Form of TGF-β1 in CHC Patients

Cat B and PKa are proteases involved in the activation of latent TGF-β1 to its active form [24,25,26,27]. TGF-β is a “master regulator” of hepatic fibrosis; it drives all stages of disease progress, from initial hepatic injury through inflammation and fibrosis to cirrhosis and hepatocellular carcinoma [28]. High levels of TGF-β activate HSCs, which trigger hepatic fibrosis [13,29]. Serum levels of total TGF-β1 and active TGF-β1 of CHC patients were significantly higher than those of HCs (*p* < 0.00001) (Figure 4A,B). These data showed an intrinsic relationship between the active form of TGF-β1 and Cat B in the serum (Figure 4C). Moreover, the levels of active TGF-β increased with the METAVIR stage of hepatic fibrosis (Figure 4D). Serum levels of the active form of TGF-β1 (%) could also be used to separate HCs (1.54 ± 0.16%), F2 (3.10 ± 0.17%), F2 from F3 (5.33 ± 0.53%), and F3 from F4 (8.32 ± 1.59%).

## 4. Discussion

Identification of the stage of hepatic fibrosis is crucial in CHC patients to initiate treatment or monitor the development of hepatic fibrosis [1,2,3]. Liver biopsy is the gold standard for staging hepatic fibrosis, but some complications limit its clinical application [30]. Thus, alternative methods and indices have been developed to estimate fibrosis/cirrhosis in the liver to substitute (at least in part) for liver biopsy.

Recent studies have demonstrated that hepatic elastography has good accuracy in predicting moderate-to-advanced hepatic fibrosis in patients with viral hepatitis [1,2,3,31]. However, because of the cost of hepatic elastography, other approaches, such as APRI and FIB4, are being used because they are based on commonly used and readily available clinical parameters [18,32,33]. The diagnostic capacity of APRI and FIB4 is controversial. They have been shown to have moderate accuracy/non-capacity to predict the initial stages of hepatic fibrosis (<F3) [33]. Our data showed that APRI and FIB4 had a low diagnostic capacity to predict F2 fibrosis, indicating that these indices alone are moderate biomarkers for predicting hepatic fibrosis in patients with HCV infection. These two indices can discriminate between the very advanced stages of hepatic fibrosis (≥F3) in CHC patients [18,32,33].

We showed that the serum levels of Cat B and PKa could be used as biomarkers for the diagnosis of hepatic fibrosis. Serum levels of Cat B were extremely increased in CHC patients with fibrosis and inflammation of the liver. Serum levels of Cat B demonstrated high discriminative capacity for the early stages of hepatic fibrosis (CHC patients with METAVIR score ≥ A1F2). A cutoff value of Cat B level (stipulated by the ROC curve) ≥ 2.61 U/mL could be used to correctly discriminate 96% of HCV samples with hepatic fibrosis (≥A1F2), as well as 96% of HCs, presenting excellent diagnostic accuracy (96%). Thus, the serum level of Cat B can be used to detect initial fibrosis and inflammation in the livers of CHC patients. This same cutoff (≤2.61 U/mL) could be used to identify 99% of HCs correctly with a high NPV, showing that measuring the concentration of Cat B in serum is an excellent diagnostic test to exclude patients with hepatic fibrosis resulting from HCV infection in the population.

The AUROC of PKa revealed a high discriminative capacity for initial hepatic fibrosis (≥A1F2). These data suggest that PKa is an excellent biomarker for predicting early hepatic fibrosis in patients infected with HCV. A cutoff ≥ 2.9 U/mL could be used to correctly identify 98% of CHC patients with hepatic fibrosis. In addition, PKa levels < 2.9 U/mL could be used to correctly identify 97% of HCs without hepatic fibrosis. PKa levels < 2.9 U/mL proved to be an excellent test to exclude hepatic fibrosis in the population.

Our data show that the serum level of cathepsin B is only a moderate biomarker to detect minimal fibrosis in F1 patients (AUROC = 0.68 ± 0.09), but is an excellent biomarker to detect intermediate levels of fibrosis in F2 patients (AUROC = 0.992 ± 0.006), and in advanced liver fibrosis in F3 (AUROC = 0.98 ± 0.01) and F4 patients (AUROC = 0.95 ± 0.02). Taken together, these results show that cathepsin B was significantly superior to APRI (AUROC = 0.71 ± 0.05) and FIB4 (AUROC = 0.67 ± 0.06) in assessing significant liver fibrosis (control versus F2–4) in CHC patients.

The diagnostic performance of serum cathepsin B and PKa for assessing significant liver fibrosis (≥F2) has proved superior to other serum biomarkers previously described [34], such as type IV collagen (AUROC = 0.73–0.83), hyaluronic acid (AUROC = 0.82–0.92), laminin (AUROC= 0.542–0.82) and Mac-2 binding protein glycan isomer, M2BPGi (AUROC = 0.774) [35].

HSCs are considered fundamental in liver response to different types of lesions, including those caused by HCV infection [36]. TGF-β1 release during hepatic injury by hepatocytes is one of the first signals for the differentiation of quiescent HSC cells and, consequently, the beginning of fibrogenesis [37]. Interestingly, cathepsin B can activate the latent form of TGF-β1 [24]. We showed that the active forms of TGF-β1 correlates with liver fibrosis grade in CHC patients, as previously indicated [38]. The active form of TGF-β1 correlated with the severity of hepatic fibrosis in CHC patients.

Manchanda et al. showed that rats with hepatic fibrosis induced by carbon tetrachloride had an increase in the hepatic expression of Cat B during fibrogenesis [12]. Furthermore, the hepatic injury/fibrosis observed during cholestasis is attenuated by the inhibition of Cat-B expression [27]. High expression of Cat B has been observed in histology sections of liver tissue of patients with hepatic steatosis and alcohol-induced hepatic cirrhosis [12].

Interestingly, cathepsin B can mediate NLRP3 inflammasome activation [39]. Moreover, inflammatory cytokines induce Cat-B secretion in the JAK/STAT-dependent signaling pathway [40]. High levels of Cat B have been observed in several pathophysiologic states involving chronic inflammation [41]. Our results corroborate those data because we observed a significant association between the severity of hepatic inflammation (≥A1) with Cathepsin-B levels in the serum of CHC patients.

PKa is strongly involved in several pathophysiologic processes associated with inflammation because it promotes bradykinin formation [42]. We showed that levels of PKa were also associated with the stage of hepatic inflammation in CHC patients. Scholars have shown that a high circulating concentration of PKa during hepatic fibrosis can be related to activation of the latent form of TGF-β1 which, consequently, leads to greater differentiation of HSCs [41,42]. PKa levels correlated strongly with Cat-B levels in the serum of CHC patients but not in HCs, suggesting that levels of both proteases are involved intrinsically in fibrogenesis in CHC patients. Previously, we demonstrated that lysosomal cysteine proteases, especially Cat B and Cat L, are involved in the activation of the zymogen form of PKa [43,44]. Taken together, these data indicate that Cat B and PKa can drive the inflammatory process, triggering hepatic fibrosis (≥A1F2) in CHC patients.

Noninvasive new Cat B and PKa methods to detect liver fibrosis may be clinically helpful in assessing the stage of fibrosis in patients with no clear indication for a liver biopsy, such as patients with chronic hepatitis B (CHB) and persistently normal serum alanine aminotransferase (ALT), patients with CHC or CHB who require follow-up assessment of the stage of fibrosis during or after treatment [45,46], and autoimmune hepatitis patients who require assessment after prolonged immunosuppressive therapy [47]. The rapid development of new medications for the treatment of some liver diseases, such as CHB, CHC and nonalcoholic fatty liver disease (NAFLD), increases the requirement for a more frequent evaluation of liver fibrosis to assess treatment response. Liver biopsies are not ideal for frequent evaluations. Cat B and PKa endopeptidase activities in the serum of patients and HCs can be measured by hydrolysis of the fluorogenic substrate in the presence or absence of specific inhibitors of the proteolytic enzymes. The emission of fluorescence intensity can be easily quantified fluorometrically using a microplate reader. This methodology is readily available, reliable, inexpensive, safe and can be well validated in different forms of chronic liver disease.

However, our study has important limitations. In addition to the small cohort, the data were obtained from CHC patients from a single center. Further studies are necessary to better characterize and validate the role of Cat B and PKa in other liver fibrosis diseases. Although our studies evaluated the accuracy of CatB and PKa in all CHC patients using liver biopsy as the gold standard reference, our protocol can also have limitations because even the best liver biopsy retains a risk of sampling error. We don’t know if Cat B and PKa measured in serum cannot be liver-specific and can be altered by pathological conditions in other tissues. APRI and FIB4 did not prove to be good markers to distinguish the individual stages of liver fibrosis; therefore, other tests with better accuracy, such as ELF, FibroTest, transient elastography-FibroScan^®^ and acoustic radiation force impulse (ARFI), should be used to validate our results obtained with Cat B and PKa. Despite the limitations pointed out, our data strongly suggest that Cat B and Pka serum levels can be used to assess liver fibrosis progression and to predict the complications and survival of liver disease in CHC patients.

## 5. Conclusions

Cat B (cutoff = 2.6 U/mL) and PKa (cutoff = 2.9 U/mL) measured in serum can be used: (i) to discriminate between different stages of liver damage in HCV-infected patients); (ii) as biomarkers to exclude hepatic fibrosis in the population. This work could enable an inexpensive, sensitive and easy-to-perform fluorometric enzymatic automated method to be inserted into routine laboratory tests to screen and classify inflammatory/fibrotic hepatic lesions (≥A1F2).

## Figures and Tables

**Figure 1 microorganisms-10-01769-f001:**
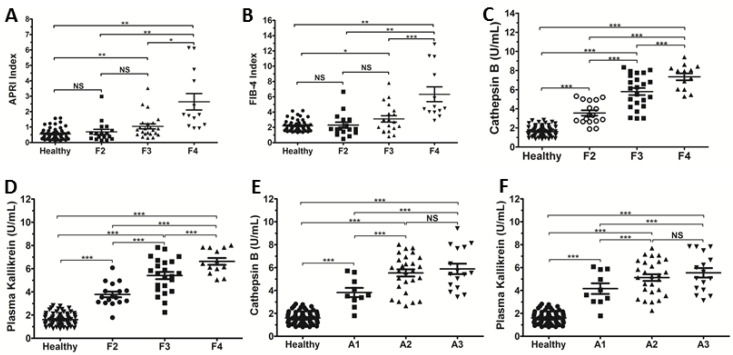
Relationship between serum levels of cathepsin B (Cat B) and plasma kallikrein (PKa) with the METAVIR scoring system for hepatic fibrosis and inflammatory activity in CHC patients and healthy controls (HCs). (**A**) APRI score in relation to hepatic fibrosis in patients with HCV infection. APRI of HCs (0.55 ± 0.05; N = 71), F2 (0.69 ± 0.17; N = 17), F3 (1.04 ± 0.16; N = 23), F4 (2.64 ± 0.53; N = 13). (**B**) FIB4 score in relation to hepatic fibrosis in patients with HCV infection. FIB4 score of HCs (2.21 ± 0.09; N = 71), F2 (2.30 ± 0.39; N = 17), F3 (3.11 ± 0.41; N = 23), F4 (6.33 ± 0.96; N = 13). (**C**) Serum Cat-B levels in relation to hepatic fibrosis in CHC patients. Cat-B levels of HCs (1.61 ± 0.06 U/mL; N = 71), F2 (3.61 ± 0.26 U/mL; N = 23), F3 (5.30 ± 0.33 U/mL; N = 30), F4 (7.14 ± 0.31 U/mL; N = 17). (**D**) PKa levels in relation to hepatic fibrosis in CHC patients. PKa levels of HCs (1.61 ± 0.06 U/mL; N = 71), F2 (3.77 ± 0.24 U/mL; N = 17), F3 (5.40 ± 0.31 U/mL; N = 23), F4 (6.62 ± 0.30 U/mL; N = 13). (**E**) Serum Cat-B levels in relation to inflammatory activity in CHC patients. Cat-B levels of HCs (1.61 ± 0.06 U/mL; N = 71), A1 (3.82 ± 0.40 U/mL; N = 10), A2 (5.26 ± 0.31 U/mL; N = 27), A3 (5.88 ± 0.46 U/mL; N = 16). (**F**) PKa levels in relation to inflammatory activity in CHC patients. PKa levels of HCs (1.61 ± 0.06 U/mL; N = 71), A1 (4.16 ± 0.46 U/mL; N = 10), A2 (5.12 ± 0.29 U/mL; N = 27), A3 (5.54 ± 0.41 U/mL; N = 16). The results are the mean ± SEM. Statistical analyses of data were conducted using ANOVA, NS = not significant; * *p* < 0.01; ** *p* < 0.001; *** *p* < 0.0001.

**Figure 2 microorganisms-10-01769-f002:**
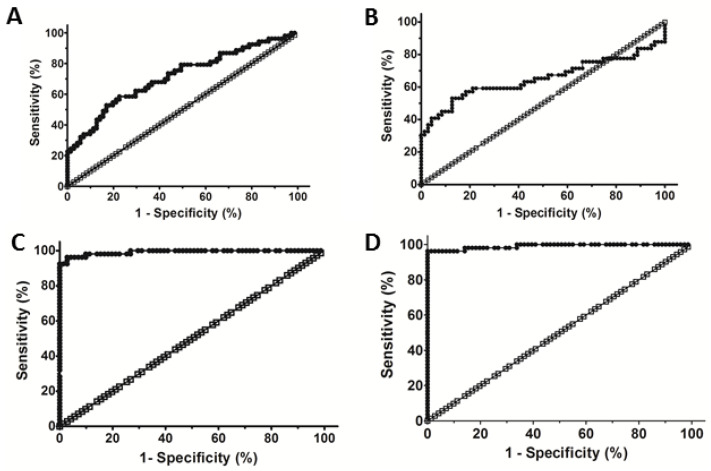
Diagnostic accuracy of APRI, FIB4 and levels of cathepsin B (Cat B) and plasma kallikrein (PKa) in CHC patients. Receiver-operating characteristic (ROC) curves were created to determine the AUROC using APRI, FIB-4 and levels of Cat B and PKa to predict hepatic fibrosis. (**A**) Evaluation of the diagnostic capacity of the APRI. (AUROC = 0.71 ± 0.05; 95%CI = 0.621–0.807; *p* < 0.0001; N = 124). (**B**) Evaluation of the diagnostic capacity of FIB4. (AUROC = 0.67 ± 0.06; 95%CI = 0.525–0.817; *p* < 0.0001; N = 124). (**C**) Evaluation of the diagnostic performance of Cat B. (AUROC = 0.99 ± 0.02; 95%CI = 0.9803–1.003; *p* < 0.0001; N = 124). (**D**) Evaluation of the diagnostic performance of PKa. (AUROC = 0.991 ± 0.007; 95%CI = 0.9770–1.005; *p* < 0.0001; N = 124). The white symbols correspond to diagonal of roc curve, where sensibility is equal to specificity, that means everyone has the same probability (50%).

**Figure 3 microorganisms-10-01769-f003:**
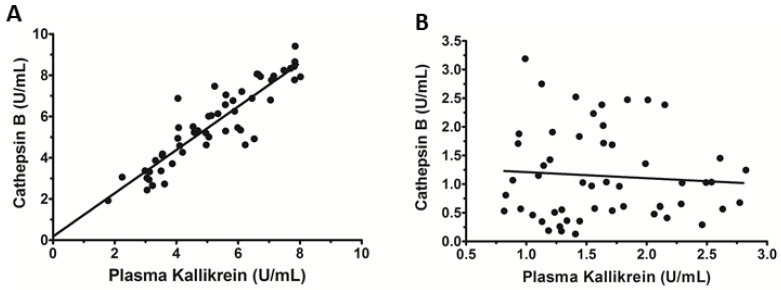
Correlation between cathepsin B (Cat B) and plasma kallikrein (PKa) levels in the serum. Statistical analyses of the data were done using Pearson’s correlation coefficient. (**A**) CHC patients with hepatic fibrosis ≥A1F2: A strong positive correlation was observed between Cat-B and PKa levels in the serum samples of CHC patients with hepatic fibrosis, Pearson’s r = 0.903; R^2^ = 0.816; 95%CI = 0.837–0.943 (*p* < 0.0001; N = 53). (**B**) Healt Controls: Pearson’s r = 0.198; R^2^ = 0.016; 95%CI = 0.149–0.384 (*p* = 0.365; N = 71).

**Figure 4 microorganisms-10-01769-f004:**
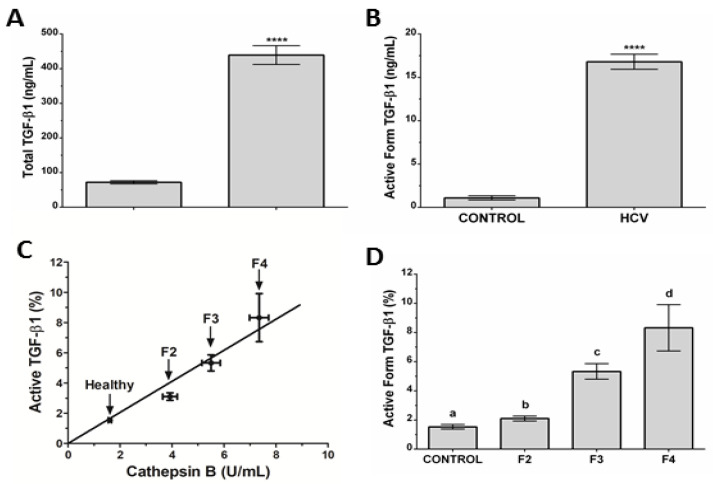
TGF-β1 levels in the serums of CHC patients and HCs. (**A**) Total TGF-β1 in HCV-infected patients and HCs. HCs (72.0 ± 4.4 ng/mL; N = 71), HCV (439 ± 27 ng/mL; N = 53). (**B**) Active form of TGF-β1 in HCV-infected patients and HCs. HCs (1.10 ± 0.25 ng/mL; N = 71), HCV (16.8 ± 0.9 ng/mL; N = 53). (**C**) Association between Cat B and the active form of TGF-β1 and hepatic fibrosis (**D**) Percentage of the active form of TGF-β1 in HCV-infected patients and HCs. HCs (1.54 ± 0.16%; N = 71), HCV (F2 = 2.10 ± 0.17%; N = 17; F3 = 5.33 ± 0.53%; N = 23; F4 = 8.32 ± 1.59%; N = 13). Results are the mean ± SEM or percentage (%) of total concentration. Student’s *t*-tests were applied to identify differences between groups and repeated-measures ANOVA with Tukey’s post-hoc test. **** *p* < 0.0001; a ≠ b ≠ c ≠ d.

**Table 1 microorganisms-10-01769-t001:** Baseline Characteristics of HCV Patients.

	CHC (N = 53)	References Value
Age (years)	57.24 ± 12.12	-
Male/Female	22/31	-
Viral Genotype		
1	84.9% (45)	-
3	15.1% (8)	-
Laboratory Date		
TB (mg/dL)	0.72 ± 0.41	<1.0
IB (mg/dL)	0.49 ± 0.28	<0.7
DB (mg/dL)	0.23 ± 0.18	<0.3
ALT (U/L)	56.64 ± 40.45	14–59 (woman); 16–63 (men)
AST (U/L)	76.98 ± 62.39	15–37
ALP (U/L)	88.64 ± 31.34	46–116
GGT (U/L)	120.89 ± 134.44	5–55 (women); 15–85 (men)
PT-activity (%)	76 ± 15	79–122
PTT (s)	30.9 ± 5.1	24.5–32.8
INR	1.21 ± 0.39	1.00–1.40
Leukocytes (10^3^/mm)	5.63 ± 1.92	4.5–10.5
Erythrocytes (10^3^/mm)	4.53 ± 0.55	4.205.40
Platelet count (10^3^/mm)	168.04 ± 84.08	150–400
Stage of Liver Disease		
F2	32.08% (17)	-
F3	43.40% (23)	-
F4	24.53% (13)	-
Grade of Inflammation		
A1	18.87% (10)	-
A2	50.94% (27)	-
A3	30.19% (16)	-
APRI	1.32 ± 1.36	<0.5 can exclude cirrhosis
		0.7 can predict significant hepatic fibrosis
		>1.0 can predict cirrhosis
		>1.5 can include cirrhosis
FIB4	3.70 ± 2.79	>3.25 suggests advanced fibrossis

Data are the mean ± SD or percentage (%). ALT = alanine aminotransferase; ALP = alkaline phosphatase; AST = aspartate aminotransferase; TB = total bilirubin; IB = indirect bilirubin; DB = direct bilirubin; GGT = gamma-glutamyl transferase; PTT = partial thromboplastin time; PT = prothrombin time; INR = international normalized ratio; APRI = aspartate aminotransferase:platelet ratio index; FIB4 = Fibrosis-4.

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
