# Peer review of "Cathepsin B and Plasma Kallikrein Are Reliable Biomarkers to Discriminate Clinically Significant Hepatic Fibrosis in Patients with Chronic Hepatitis-C Infection"

_microorganisms, 2022, doi:10.3390/microorganisms10091769_

Round 1

Reviewer 1 Report

1.Patient number was small. It was not a useful marker to predict the fibrosis

2. How about the AUROC curve to predict the cirrhosis ?

3. What is the clinical characteristic between hepatic fibrosis (or cirrhosis) and healthy control?

Author Response

  1. Patient number was small. It was not a useful marker to predict the fibrosis

Answer: Our cohort, although small, is very consistent because our patients are totally controlled. The liver fibrosis and inflammation parameters were obtained from liver biopsies in all CHC patients (N=53), that is the gold-standard method for measuring of hepatic fibrosis and inflammation in HCV-infected patients. We also excluded patients with other liver diseases that cause hepatic fibrosis. All CHC patients were clinically compensated, under outpatient medical follow-up. Our data shows that in these controlled cohort, that the cathepsin B and Kallikrein serum levels were more efficient to detected fibrosis than estimated equations APRI and FIB 4 indexes, that are normally used for in accomplished patients with hepatic commitment.

  1. How about the AUROC curve to predict the cirrhosis?

Answer: As suggested by Reviewer 1, the ROC curve was done to predict the liver cirrhosis and its area under ROC curve (AUROC) was calculated. The diagnosis performance of cathepsin B to predict cirrhosis in CHC patients is also very powerful (F4 AUROC = 0.95 ± 0.02). Please, see these data in the revised manuscript at the page 8, first paragraph, lines 263-273.

  1. What is the clinical characteristic between hepatic fibrosis (or cirrhosis) and healthy control?

Answer: The clinical laboratory data of all CHC patients revealed no jaundice (with bilirubin values within reference ranges) and showed alterations in laboratory values compatible with CHC infection. Distributions for fibrosis stage and the inflammatory activity grade in the study population are also presented in Table 1. If the mean APRI score was >1.5, CHC patients had hepatic cirrhosis. As expected, HCs (blood donors) had no hepatic fibrosis. FIB4 scores in the CHC group suggested advanced fibrosis (>3.25). Normal APRI and FIB4 scores were found in HCs.

Reviewer 2 Report

Alexia et al. share an intresting work concerning three direct fibrosis markers in HCV patients' serum. Two of them seem to be particularly useful (Cathepsin B, Kallikrein) in prediction in fibrosis prediction.

Nevertheless, I have some remarks:

- It woild be interesting to compare them with advaced tests, such as FibroTest or FibroScan, rather than APRI or FIb4.

- AUROC is interesting to separate fibrosis/no fibrosis, but It would be more interestinf to separate each stage.

- Discussion must be improved. Limitations were not discussed (small sample, unicenter, .....). and possibilities in other chronic liver diseases (NASH, ....).

Author Response

#Reviewer 2

Alexia et al. share an interesting work concerning three direct fibrosis markers in HCV patients' serum. Two of them seem to be particularly useful (Cathepsin B, Kallikrein) in prediction in fibrosis prediction.

Nevertheless, I have some remarks:

- It would be interesting to compare them with advanced tests, such as FibroTest or FibroScan, rather than APRI or FIb4.

Answer: We Thank for these Reviewer suggestions. But in our cohort, all CHC patients (N=53) were biopsied as gold-standard test, Fibroscan were not performed because it was not disponible for us use during the development this work. But we will consider this pointin in our posterior study.

- AUROC is interesting to separate fibrosis/no fibrosis, but It would be more interesting to separate each stage.

Answer: As suggested by Reviewer 2, the AUROC of cathepsin B was also calculated for each grade of liver fibrosis. We observed the diagnosis power of the cathepsin B to predict liver fibrosis in function of the METAVIR score: Control versus F2+F3+F4, AUROC = 0.992 ± 0.006 (95%CI = 0.98–1,00; P < 0.0001), Control+F2 versus F3+F4 AUROC = 0.98 ± 0.01 (95%CI = 0.96–1.01; P < 0.0001) and Control+F2+F3 versus F4 AUROC = 0.95 ± 0.02 (95%CI = 0.91–0.99; P < 0.0001). We also determined the diagnosis power of cathepsin B in F1 patients, which were not previously classified as CHC patients. Cathepsin B only displayed a moderate diagnostic capacity to detect fibrosis in F1 patients, AUROC = 0.68 ± 0.09 (95%CI = 0.51–0.86; P = 0.035), but cathepsin B also showed an excellent diagnosis power to detect intermediate levels of fibrosis in F2 patients (AUROC = 0.992 ± 0.006), and advanced liver fibrosis in F3 (AUROC = 0.98 ± 0.01) and F4 patients (AUROC = 0.95 ± 0.02). Please, see these data in the Results Section of revised manuscript version at the page8, first paragraph, lines 263-273.

- Discussion must be improved. Limitations were not discussed (small sample, unicenter...). and possibilities in other chronic liver diseases (NASH, ....).

Answer: The authors thank the Reviewer 2 for these observations. The presence of another chronic liver diseases was excluded, these information were included in Materials and Methods Section of the reviser manuscript at page 2, last paragraph, lines 90-95. The major limitations of this work were discussed in Discussion Section of the revised manuscript at the of pages 10, last paragraph, lines 382-406.

Reviewer 3 Report

Zanelatto, et al. showed that higher serum levels of cathepsin B (Cat B) and plasma kallikrein (PKa) were observed in CHC patients with severer hepatic fibrosis, using liver biopsy specimens. They also showed that Cat B and PKa were superior to APRI or FIB4 in the diagnosis of liver fibrosis, claiming that these could be novel biomarkers for liver fibrosis. They also demonstrated that serum Cat B levels were well-correlated with serum TGF-b1 level. These data are interesting and could be clinically valuable, but some issues remain to be addressed.

Major

1.       I guess these CHC patients would be treatment-naïve, but the authors should clearly describe it.

2.       Serum levels of Cat B and PKa were well-correlated with hepatic fibrosis as well as hepatic inflammation. I am wondering if Cat B and PKa are fibrosis markers or inflammation markers. Measuring serum Cat B and PKa after treatment with DAA could answer this question.

3.       Authors should discuss the superiority of Cat B and PKa to other hepatic fibrosis markers such as hyaluronic acid, type IV collagen, TGF-b1, and M2BPGi.

Minor

Line 59: “can the degrade””can be the degrade”?   

Author Response

#Reviewer 3

  1. I guess these CHC patients would be treatment-naïve, but the authors should clearly describe it.

Answer: We thank Reviewer 3 for these observation. Yes, all CHC patients were treatment-naïve, this information was inserted in revised text in Methodology Section, page 3, lines 123,124.

  1. Serum levels of Cat B and PKa were well-correlated with hepatic fibrosis as well as hepatic inflammation. I am wondering if Cat B and PKa are fibrosis markers or inflammation markers. Measuring serum Cat B and PKa after treatment with DAA could answer this question.

Answer: We thank the Reviewer for this is excellent observation. Cathepsin B and plasma kallikrein are involved in several inflammation states as in pathogenesis of liver fibrosis, including directly proteolytic activation of TGF-β1, please see these comments in Discussion Section of the revised manuscript, pages 10-11, lines 353-396. In this context, Cat B and PKa may rise as potential biomarkers of HCV-infected patients. The role of Cat B and PKa in inflammation state have been studied and, although this present work can dissociate both liver fibrosis and liver inflammation in CHC patients. Yes, we totally agree with you that measuring serum Cat B and PKa after treatment with DAA could answer this question.

  1. Authors should discuss the superiority of Cat B and PKa to other hepatic fibrosis markers such as hyaluronic acid, type IV collagen, TGF-b1, and M2BPGi.

Answer: The diagnostic performance of serum cathepsin B and PKa for assessing significant liver fibrosis (≥F2) has proved superior to other serum biomarker previously described [34], such as: type IV collagen (AUROC = 0.73-0.83), hyaluronic acid (AUROC = 0.82–0.92), laminin (AUROC= 0.542–0.82) and Mac-2 binding protein glycan isomer, M2BPGi (AUROC=0.774) [35]. please see these comments in Discussion Section of the revised manuscript, page 10, 3º paragraph, lines 348-352.

Round 2

Reviewer 1 Report

No further comments

Author Response

No further comments

Answer: We thanks the comments and analysis made by Reviewer1, they contributed to the improvement of our work.

Reviewer 2 Report

I thank the authors for their reply and the modifications in mansucript.

But, sorry, APRI and Fib4 are not good fibrosis markers. I can understand that other tests like FibroScan or serum complex tests such as ELF, FibroTest, FibroMeter. They are more accurate than APRI and Fib4 to distinguish fibrosis stages.  I think that the authors must explain this obstacle in their limitations.

Author Response

But, sorry, APRI and Fib4 are not good fibrosis markers. I can understand that other tests like FibroScan or serum complex tests such as ELF, FibroTest, FibroMeter. They are more accurate than APRI and Fib4 to distinguish fibrosis stages.  I think that the authors must explain this obstacle in their limitations.

Answer:  We fully agree with the comments made by Reviewer2.  Indeed, APRI and FIB4 did not prove to be good markers to distinguish the individual stages of liver fibrosis, therefore other tests with better accuracy, such as: ELF, FibroTest, transient elastography-FibroScan® and acoustic radiation force impulse (ARFI) should be used to validate our results obtained with Cat B and PKa.

As suggested by Reviewer2, these comments were inserted in the Discussion Section of the revised manuscript in page 11, first paragrap, lines 405-408.